# 2-Piece Cork Stoppers as Alternative for Valorization of Thin Cork Planks: Analysis by LCA Methodology

**DOI:** 10.3390/foods10040873

**Published:** 2021-04-16

**Authors:** Francisco Javier Flor-Montalvo, Agustín Sánchez-Toledo Ledesma, Eduardo Martínez Cámara, Emilio Jiménez-Macías, Jorge Luis García-Alcaraz, Julio Blanco-Fernandez

**Affiliations:** 1Higher School of Engineering and Technology, International University of La Rioja (UNIR), 26004 Logroño, La Rioja, Spain; agustin.sancheztoledo@unir.net; 2Department of Mechanical Engineering, University of La Rioja, Luis de Ulloa 20, 26004 Logroño, La Rioja, Spain; eduardo.martinezc@unirioja.es (E.M.C.); julio.blanco@unirioja.es (J.B.-F.); 3Department of Electrical Engineering, University of La Rioja, Luis de Ulloa 20, 26004 Logroño, La Rioja, Spain; Emilio.jimenez@unirioja.es; 4Department of Industrial Engineering and Manufacturing, Autonomous University of Ciudad Juarez, Ciudad Juárez CHIH 32315, Mexico; jorge.garcia@uacj.mx

**Keywords:** life cycle assessment, cork stoppers, thin cork planks, valorization

## Abstract

Natural stoppers are a magnificent closure for the production of aging wines and unique wines, whose application is limited by the availability of raw materials and more specifically of cork sheets of different thickness and quality. The growing demand for quality wine bottle closures leads to the search for alternative stopper production. The two-piece stopper is an alternative since it uses non-usable plates in a conventional way for the production of quality caps. The present study has analyzed the impact of the manufacture of these two-piece stoppers using different methodologies and for different dimensions by developing an LCA (Life Cycle Assessment), concluding that the process phases of the plate, its boiling, and its stabilization, are the phases with the greatest impact. Likewise, it is detected that the impacts in all phases are relatively similar (for one kg of net cork produced), although the volumetric difference between these stoppers represents a significant difference in impacts for each unit produced.

## 1. Introduction

### 1.1. The Cork Oak

The cork oak is a tree that belongs to the *Quercus* genus; hence, its name *Quercus suber* L. It is a tree with green leaves throughout the year that, for an adult specimen, has an average height of between 10 and 15 m and a diameter between 0.3 and 1 m [1,2]

The cork oak is a species associated with the Mediterranean Sea. The climate is dry in summer, with high temperatures and little rainfall, while winters have mild temperatures and abundant rainfall, although they tend to be irregular, alternating periods of great abundance of water with others of drought [3]. 

Within the Mediterranean climate zones, the cork oak is more common in those areas with a great oceanic influence that favors a damping of both the effects of drought and temperature. This fact, combined with the great light needs of the cork oak, determine the presence of this arboreal species in very specific areas. Likewise, it is a species that does not proliferate outside the areas and conditions outlined, which makes cork a unique, scarce and valuable product [4]. In these forests it can be found both as a min tree species and together with other Mediterranean trees [5].

The bark of the cork oak is very thick, grayish in color, and usually cracked. During the “saca” or extraction of the cork, the trunk without the bark presents a reddish-yellowish color [6,7]. The main use of the cork oak is the product known as cork. Cork is the bark of the cork oak and has peculiar characteristics, among which are its lightness, flexibility, high coefficient of friction, impermeability, which, provided that it is natural, recyclable, and renewable, makes it a very interesting material for various applications [7,8].

### 1.2. The Cork Manufacture

Modern cork manufacture dates back to the mid-18th century. Initially, and until the end of the 19th century, it was a business destined to the wine sector and more specifically to the manufacture of cork stoppers for still and sparkling wines.

The stoppers manufactured were in any case natural stoppers made from cork sheet. At the beginning of the 20th century, and faced with the impossibility of meeting the demands for stoppers for closing wines and sparkling wines, new products appeared, among which the agglomerated cork stopper stands out. The main advantage of these stoppers was the possibility of manufacturing them from cork sheets that, due to their thickness and geometry, could not be used to obtain natural stoppers, which allowed increasing the number of stoppers available on the market.

The effect of the appearance of the agglomerate had two immediate consequences in the productive manufacturing structures. In the first place, the great cork transforming powers during the 19th century, Spain and to a lesser extent Portugal, reduced their weight with the entry of new competitors. This led to a reindustrialization of the cork sector in Spain and to a lesser extent in Portugal [9,10]. On the other hand, the demand for cork for non-oenological uses, although it primarily used thin cork (also known as branch cork), it also diverted part of the cork traditionally used for the manufacture of natural stoppers, reducing the availability of this type of stoppers [11]. Currently, Spain and Portugal account for more than 80% of the cork extracted (Portugal exports 62.4% of all the cork in the world, followed by Spain with 18.5% according to 2017 data) and more than 60% of the world’s cork oak area available.

Although in the last 4 years the vineyard area in the world has experienced a decrease, the wine area in 2018 stabilized at around 7.4 million hectares and the volume of exports from all countries reached a volume of 108.0 M hL and a value of EUR 31 300 M. Of these, bottled wines accounted for 53% of the total international wine trade by volume, although these bottled wines represent 70% of all exported wines in terms of total economic value [11].

Based on the foregoing, it can be deduced how the market is being segmented into wines of increasing quality and usually associated with careful and/or unique aging, while the weight of lower added value wines, usually associated with the bulk sales, decreases.

### 1.3. The Natural Stopper and the Two-Piece Natural Stopper

Based on the data presented, it is evident that, although the natural stopper is a magnificent closure for the production of aging wines and singular wines, only those stoppers of high quality and usually associated with wines of medium and high categories are accepted by the current market.

There are serious limitations regarding the total annual cork production and its recurrence that, together with the development of new substitute products for both cork (agglomerate, technical stopper, microgranulates, etc.) and other materials (polymeric closures, screw cap, etc.) has made many products migrate to this type of closures.

In any case, and in the face of a growing market for quality and bottled wines, the demand for high-end stoppers is greater, although the supply cannot be increased.

Nor would an adequate forestry policy allow a substantial increase in the amount of cork available in the medium term due to the long maturation period of the cork oak forest before producing its first extraction of cork useful for the wine sector.

Under these circumstances, the opportunity arises to manufacture two-piece stoppers from cork sheets that are less thick and which cannot be used to make one-piece stoppers.

A two-piece stopper is a stopper made up of two semi-cylindrical halves glued together in the middle.

It maintains the main characteristics of the natural one-piece stopper, at a slightly lower cost and with some additional advantages:Greater use of raw materials in the production of high added value caps.Less expensive raw materials and the absence of supply problems when competing for their acquisition with other products with lower added value.Greater availability of raw materials in the market by competing only with low value-added processes (cork rolling and cork crushing to agglomerate/microgranulate).Branch cork, since it is far from the traditional sources of cork contamination (it is far from humidity, the main cause of the appearance of 2,4,6-Trichloroanisole—TCA—main cause of olfactory defects in wine associated with the use of cork stoppers), which implies lower levels of contamination than conventional cork [12].Density and mechanical capacities of the stoppers manufactured with this process, is much higher than those manufactured by conventional processes (“thin” cork has the same number of cell walls for years of growth, but since its thickness is much smaller, the number of cell walls per linear cm is much higher).When working with base plates of less thickness (22% of the thickness required for the perforation of a conventional stopper), it is much easier to treat said plates in depth using heat treatments and minimize the appearance of TCA.

However, since the manufacturing process is different from that used to produce 1-piece stoppers, the manufacturing impact of this type of two-piece stoppers will also be substantially different.

### 1.4. Research Context

There exist Life Cycle Assessments associated with the stopper manufacturing phases [13,14,15], which carry out an exhaustive analysis in order to assess the environmental impacts derived from the production of natural cork stoppers in Portugal, in addition to identifying the stages and processes to suggest improvement actions and alternative scenarios, as well as LCA environmental impact analysis for the production of natural cork stoppers [13,16], granules [17], or for champagne bottles [18]. However, the impact of this (two-piece) type of manufacturing process has not been analyzed, which, as it has been indicated, would allow the market to supply high quality stoppers with excellent properties without the limitations associated with the current availability of cork.

In order to complement the information available regarding the Life Cycle Analysis of two-piece cork stoppers, an environmental study is carried out for the manufacture of two-piece natural cork stoppers in the most common dimensions demanded by the market and independently analyzing the different production methods that exist today: drilling and turning.

All the processes will be analyzed from the reception of the raw cork sheet to the achievement of the finished and personalized cork stoppers, ready to be used in the winery to close the wines.

To obtain the primary data, 10 batches of stoppers were measured on the manufacturing process for each type of cap, analyzing diameter, and length.

In the case of caps of the same diameter, the analysis of the initial phases up to the initial cutting of the plate has been carried out jointly.

### 1.5. Justification and Object of the Research

Based on the above, it is evident that there is a growing demand for cork stoppers for aging wines at the same time that the production of quality natural stoppers does not grow in quantity and even decreases with the use of large volumes of cork for manufacturing microgranulated or agglomerated stoppers.

The increase in the availability of raw cork results, as explained above, in a very long process that involves silvicolous work for several decades.

On the other hand, a thick cork sheet is necessary to be able to produce cork stoppers, those with insufficient thickness being used for the production of agglomerated or microgranulated stoppers, for the production of washers or for other productive sectors.

Based on the foregoing, the two-piece stopper is an acceptable alternative given that it makes it possible to take advantage of the cork with less thickness and produce stoppers that are very similar to the one-piece stopper maintaining a large part of its added value. (Figure 1).

However, the production process is different and includes additional phases, so, consequently, its environmental impact will also be different, being the object of this study to determine the comparative impact of the manufacture of two-piece stoppers for different diameters and lengths, and production system.

## 2. Materials and Methods

### 2.1. Objectives and Scope

In this article, a gate to gate analysis has been carried out for 2 different geometries 24 and 26 mm diameter and 44 mm length and for 2 different production processes, turning process and drilling (perforating) process, considering all the production processes.

For each of these production processes, primary data has been obtained for each of the processes, measuring directly on each of the workstations, the consumption of materials and water, the working times, as well as the waste produced, and their destinations and use.

Real measurements have been taken for each of the processes, thus obtaining primary data on material consumption, working hours, electricity and water consumption, as well as all the other data mentioned in the document.

There are two different objectives throughout this research:-to identify differences between different processes and geometries.-to identify the critical activities of the production process.

In this way, this study will allow the detection of the critical stages of the production process and the identification of the evolution of the impact of the activity throughout the production activity.

The production process analysis required the use of Simapro 8.3 software and the CML-IA baseline V3.04/EU25 calculation methodology.

For the analysis of the production process, the Simapro 8.3 software and the CML-IA baseline V3.04/EU25 calculation methodology have been used. The following impact categories have been selected:Abiotic Deplation (AD);Global Warming-GWP100 (GW);Ozone layer depletion (OLD);Human toxicity (HT);Fresh water aquatic ecotoxicity (FWAE);Marine aquatic ecotoxicity (MAE);Terrestrial ecotoxicity (TE);Photochemical oxidation (PO);Acidification (AC);Eutrophication (EU).

### 2.2. Functional Unit

After the analysis carried out, a kg of fully processed and marked 2-piece stoppers has been defined as the functional unit for the present study, including the post-treatment and packaging processes.

### 2.3. System Limits

The production processes have been divided into three main blocks:initial operations to be carried out on raw cork sheet;machining operations to obtain raw stoppers;post-treatment and customization and packaging operations.

The limits of the system have been defined so that they include all the activities and processes necessary for the manufacture of finished stoppers and consequently to allow obtaining the functional unit. This includes all the processes necessary for the transformation of unsorted raw plates into packaged finished stoppers (see Figure 2).

Based on the above, the following activities have been considered within the limits of the system:Electricity consumption of all equipment and machines used in the production process.Labor and associated emissions.Consumption of materials throughout the production process.Consumption of other resources throughout the production process.Complete packaging of the functional unit.Recovery of by-products and cork waste for other applications with economic return.

It is specified that the following activities are specifically excluded from the production process:The transport of raw materials, whatever their origin, to the production plant.Installation and dismantling of production plant.Maintenance of machinery and process facilities.Management and treatment of waste produced in the process.Storage and transportation of finished product to customer facilities.

### 2.4. Assumptions

The following aspects will be taken into account with respect to the analyzed process:Lot characteristics: Raw cork platform. Average load 365 kg/platform.Number of batches studied: for each process analyzed and each dimension and diameter, 10 batches were considered, each consisting of a raw cork platform.Cork density: the average density of the worked cork has been 191.37 kg/m^3^,Volume of caps: 1.99051E-05 m^3^/stopper for diameter 24 mm and length 44 mm. and 2.33609E-05 m^3^/stopper for diameter 26 mm and length 44 mm.Production staff: eight-hour production shift per day and five working days per week.

### 2.5. Inventory

After a detailed analysis of the production process (Figure 3), it has been segmented into three main blocks of processes:initial operations to be carried out on raw cork sheet;machining operations to obtain raw caps;post-treatment and customization and packaging operations.

Next, Table 1, Table 2, Table 3 and Table 4 show the summary of the inventory for each of the phases and each of the types of stoppers analyzed.

The consumption in each phase has been considered for a process input of 1000 kg of raw cork. The difference between the entrance of raw cork and the consumptions of each phase will correspond to the net use destined to the manufacture of stoppers. The negative values in phase 3 correspond to the reclassification of stoppers and material initially discarded and that later is reprocessed.

#### 2.5.1. Phase 1: Initial Operations to Be Carried Out on Raw Cork Sheet

Selection of cork sheets: The raw sheets sent after the cork extraction process must be selected. Those plates that present contamination that could significantly affect the stoppers (green, pitted, earth, rot, yellow stain, etc.) will be eliminated.

Those plates that do not meet the appropriate dimensional criteria for the planned production process are also segregated.

Boiling of cork sheets: Selected raw sheets undergo a hot water boiling process with or without the addition of phenolic additives.

Additive: Sometimes they also receive a steam treatment in order to improve their organoleptic conditions.

Through this process it is possible to sanitize the plate minimizing the subsequent appearance of problems and transfers of odors and flavors while achieving a significant improvement in the mechanical properties of the treated plates by homogenizing the presence of water in the microstructure of the cork and with it its mechanical behavior. By hydrating the iron, an increase of close to 20% in its volume is achieved (once the excess supply water has been eliminated), substantially increasing its elasticity and softness.

Resting and drying: Once the plates have been cooked, they must be left to rest in order to eliminate excess moisture until reaching approximately 14% and to stabilize their structure. Although this process has a minimum duration of 15 days, depending on the characteristics of the cork and its specific state, it can take much longer, sometimes reaching several months.

Drying and resting will be carried out under cover but in an open pavilion that allows the air circulation.

Artificial drying processes are not recommended for treating cork.

#### 2.5.2. Phase 2: Machining Operations to Obtain Raw Caps

In this phase, the raw cork sheets are machined to obtain cylindrical stoppers with the required dimensions.

There are two main and different manufacturing processes that have been analyzed throughout this paper:Manufacture of perforated stoppers.Manufacture of turned stoppers.

In the case of the manufacture of perforated stoppers, the cross-cutting of cork sheets is initially carried out.

After that, the “belly” or inner surface of the cork oak bark will be removed, thus achieving a flat and polished surface, suitable for gluing.

Once this cut has been made, two pieces will be glued, keeping them in a press with pressure and temperature for 24 h.

After finishing the gluing process, the glued strips are subjected to a drilling process and subsequent operations (parting, head polishing, body polishing, selection, etc.) in a similar way to the process carried out in the manufacturing process of a 1-piece stopper.

In the case of turned plugs, the main difference is that, after obtaining strips from the raw plate, two cuts are made to each strip, eliminating the “belly” but also the “back” (outer part of the bark of the cork oak) from cork.

After that, the strips obtained are glued by means of a press with pressure and temperature input and, after 24 h of stabilization of the glue, the glued strips are extracted from the press and turned into bars.

The bars are cut into plugs using a multiple cutter, after which cylindrical plugs are obtained that are subjected to the rest of the usual processes mentioned above (head polishing, body polishing, selection, etc.).

#### 2.5.3. Phase 3: Post-Treatment and Customization and Packaging Operations

In this last block, raw stoppers with the desired size are started and they are subjected to the processes aimed at their organoleptic and aesthetic conditioning as well as their personalization.

Ink marking: Although it is possible to mark the caps with fire or ink, given that the market is increasingly inclined towards ink marking of caps, this process has been considered. Through a continuous rotary marker, the stoppers receive the desired impression on the body. The decoration of the heads is not usual since they will be in contact with the wine.

Siliconizing: In this phase, a liquid silicone film will be applied to the surface of the stopper, improving its tightness and ensuring that the stopper does not deteriorate over the years in contact with the wine.

This process is carried out in a perforated rotary drum with temperature control.

Waxing of surface: The stoppers receive a layer of paraffin in order to ensure that the removal of the stopper is adequate and does not involve excessive effort. This process is carried out in a rotating drum with temperature control.

Drying and stabilizing: once the silicone and waxing processes are finished, the caps are dried and stabilized to avoid possible biological contamination.

Packaging: The finished caps are packed in Polyethylene bags with an oxidizing gas (SO_2_) in order to avoid eventual contamination of the caps until they are used.

## 3. Results

Throughout this work, a LCA has been carried out in order to identify the main environmental impacts associated with the manufacture of two-piece natural cork stoppers from thin sheet cork.

For this, a gate-to-gate approach has been considered, thus analyzing all the environmental impacts of the production process for stoppers of 24 and 26 mm diameter and 44 mm lengths and through drilling and turning technologies for each of the cases.

To carry out the mentioned LCA, SimaPro 8.3 software was used and the CML-IA baseline V3.04/EU25 methodology was applied to calculate the environmental impact.

For the present case, the proposed functional unit has been 1 kg of fully processed and marked two-piece stoppers, including post-treatment processes and packaging.

The closure manufacturing activities have been divided into three different phases, for which all the impact categories have been determined.

To facilitate the understanding of the results obtained, the global impact of the product is first shown, and subsequently, for each of the proposed plug manufacturing geometries and methodologies, the impacts for each of the phases analyzed are presented.

### 3.1. Global Analysis of the Environmental Impact

The results obtained for each of the impact categories studied and associated with the different geometries and productive methodologies proposed can be seen in Table 5.

A detailed analysis of these data drives to affirm that the manufacturing processes of stoppers by turning generate, for all categories, greater impacts than the traditional turning system.

On the other hand, the geometries of stoppers with a diameter of 26 generate slightly lower impacts than those produced by the manufacture of stoppers with a diameter of 24 mm. However, this effect is much lighter than that associated with the different cap manufacturing methodologies.

### 3.2. Analysis of the Impacts Generated in Stage 1 for Each of the Geometries Studied

Stage 1 is composed of three processes: Cork sheets selection, Boiling of cork sheets, and Resting and Drying.

Table 6 shows the impacts generated by these activities for each of the stopper geometries under study.

Based on the previous data, it is concluded that, since the process so far is similar and the same raw plates are processed, the results obtained at this stage are very similar, yielding very similar results and only differentiated by the suitability of each piece of cork for the process and the geometries chosen.

Subsequently, once the selection of plates based on the geometry as well as their machining begins, those values will change substantially.

### 3.3. Analysis of the Impacts Generated in Stage 2 for Each of the Geometries Studied

Stage 2 in the case of turned stoppers consists of 11 processes during which the raw plate is converted into machined and finished stoppers, although not yet customized or subjected to the finishing processes.

For the present study, the manufacture of these stoppers has been analyzed using drilling technology and turning technology.

Table 7 and Table 8 show the impacts generated by this phase, breaking down the impacts of each of the activities for the manufacture of perforated stoppers of 44 mm in length and 24 and 26 mm in diameter respectively.

From the comparison of the impacts generated by the caps of 24 mm and 26 mm in diameter and 44 mm in length, a significant reduction in such impacts in all categories can be seen in this phase in those stoppers with a larger diameter.

This is due because, in the case of the perforated process, since the process analyzed works in excess thicknesses in the case of the manufacture of two-piece stoppers when working with thin plates, the raw plate and therefore the raw material necessary to obtain these stoppers is practically the same.

Despite the fact that the number of viable stoppers of 26 mm will be lower than those obtained for a diameter of 24 mm, the difference in useful volume of the cap and therefore the use of raw material and other inputs will be much greater, thus reducing the impacts generated by each kg of cork obtained.

In the case of perforated caps, stage 2 consists of 11 processes during which the raw plate is converted into machined and finished stoppers, although not yet customized or subjected to the finishing processes.

Table 9 and Table 10 show the impacts generated by this phase, breaking down the impacts of each of the activities for the manufacture of 44 mm long and 24 and 26 mm diameter perforated stoppers, respectively.

As in the previous case and for similar reasons, from the comparison of the impacts generated by the stoppers of 24 mm and 26 mm in diameter and 44 mm in length, a significant reduction in such impacts is observed at this stage in all the categories in those stoppers with a larger diameter.

However, in this case the comparison shows significantly lower differences than those obtained in the case of perforated stoppers.

The difference mainly lies in the fact that in the case of the turning process, the number of caps obtained from 1 kg of raw material is less than that obtained in the case of perforated stoppers, which dampens the effect associated with the diameter and the use of material depending on it.

### 3.4. Analysis of the Impacts Generated in Stage 3 for Each of the Geometries Studied

Stage 3, the objective of which is the application of final surface treatments, the customization of the stoppers and its packaging, is made up of 5 different processes: ink marking, silicone coating, waxing, drying, and counting and packaging.

Table 11, Table 12, Table 13 and Table 14 show the impacts generated by these activities for each of the stopper geometries under study.

From the above data, it can be deduced that the differences between the impacts generated by stoppers of equal dimensions are very small.

However, there are profound differences between stoppers of different diameters regardless of the manufacturing methodology.

This is because these processes mainly consist of the superficial application of treatments or products, so the perimeter/volume relationship is critical.

Likewise, the analysis of the packaging activity is greatly affected by the difference in the net volume of stoppers within the same container bag, since a fixed number of stoppers is included in one of these bags regardless of their geometry, resulting in an impact smaller to greater diameter or length of the plug.

## 4. Discussion

Throughout this paper, a door-to-door LCA of the complete two-piece stopper manufacturing process in 24 and 26 mm diameters and 44 mm length is presented, manufactured by both turning or drilling (perforating) techniques.

The LCA has been developed based on real data from a production plant during the manufacturing process of 10 cork platforms weighing 365 kg/platform for each of the defined geometries.

In this way, precise primary data on the environmental impact of each of the individual activities that define the production process have been obtained.

To help accurately identify the influence of each of the manufacturing stages on the impacts generated for each of the geometries and production methodologies proposed, Table 15, Table 16, Table 17 and Table 18 show the influence of each of these phases on each of the proposed study elements.

From the above results it can be deduced that, for all the manufacturing processes, stage 1, mainly due to the boiling processes with a considerable consumption of water and energy derived from the heating for boiling the cork sheets, is the one that generates greater impacts.

Stages 2 and 3, although have an influence on the impacts generated, they are much lower.

It is also important to note that the impacts of phase 1 are very similar between the different proposed production methodologies and identical between different geometries manufactured with a single production methodology.

From these data, it can be deduced that there is an important relationship between the volume of use of the raw cork sheets and the impacts generated by the activity since, as it is not possible to classify the sheets until the end of their stabilization process, independently of the selected geometry there will be a consumption of resources directly associated with the processing of each unit of raw cork.

On the other hand, the difference in impacts in phase 1 associated with the turning or drilling processes lies in the different input geometries used which, ultimately, imply slight changes in the consumption of water and energy for boiling the cork.

Throughout this research, a kg of fully processed and marked two-piece stoppers has been defined as the functional unit for the present study, including the post-treatment and packaging processes.

Although this presentation of the results allows them to be analyzed in a homogeneous way and allows their application from the point of view of all the actors in the value chain of the product, it makes their interpretation difficult for wine producers, since a transformation is necessary for each of the geometries shown.

To this effect, Table 19, Table 20, Table 21 and Table 22 show the global impacts for a unit of cork stopper of the chosen geometry.

## 5. Conclusions

Throughout the present study, a gate to gate LCA of the complete manufacturing process of two-piece stoppers in diameters 24 and 26 mm and 44 mm in length has been carried out, manufactured by both turning or drilling methodologies.

Through this work and attending to first objective of the research, it has been determined that stage 1, corresponding to the initial processing procedures of the plate, its boiling and stabilization, is the one that generates the greatest impacts, well above stage 2, corresponding to the machining processes of the stopper from the plate, or from phase 3, corresponding to the post-treatment and personalized processes of the stopper and its packaging for delivery to the customer.

The comparative analysis between different production methodologies and stopper geometries and attending to second objective of the research, shows many similarities in phase 1 because the treatment process of the cork sheet is analogous. However, it must be emphasized that the data are not the same because the caliber of the plates necessary for the manufacture of 26 mm stopper is greater than in the case of 24 mm stopper; therefore, for the same initial weight, the number of plates is lower and consequently the consumption is also lower.

As a conclusion to the above, a relationship is established between the volume of use of the raw cork sheets and the impacts generated by the activity.

Considering the impacts generated by the machining and post-treatment processes of the stoppers, it can be concluded that although the impact generated for different methodologies and for different diameters is similar for each kg of net cork produced, the impact of the larger diameter stoppers will in turn also be much greater because the volume of such stoppers is also considerably higher (more precisely 17.36% higher).

## Figures and Tables

**Figure 1 foods-10-00873-f001:**
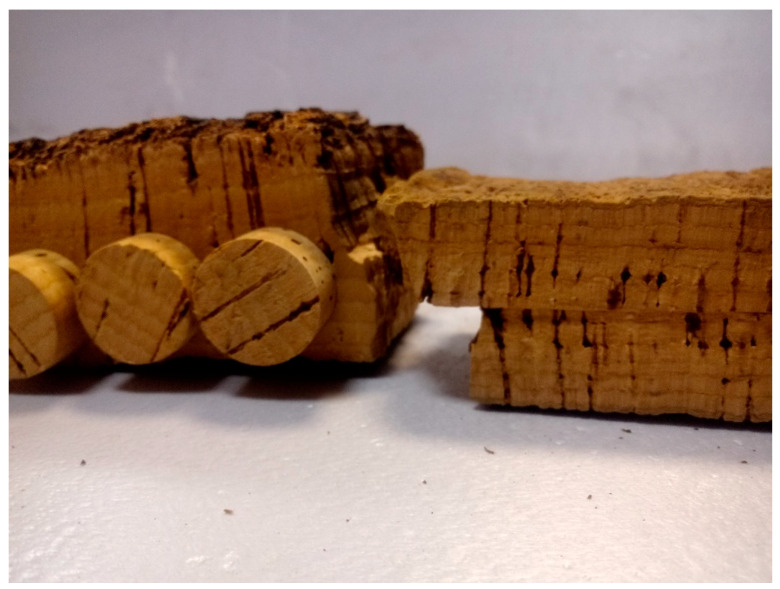
Perforated cork sheet to obtain one-piece stoppers on the left and 2 thin cork sheets glued together to achieve the same useful thickness and manufacture two-piece stoppers on the right.

**Figure 2 foods-10-00873-f002:**
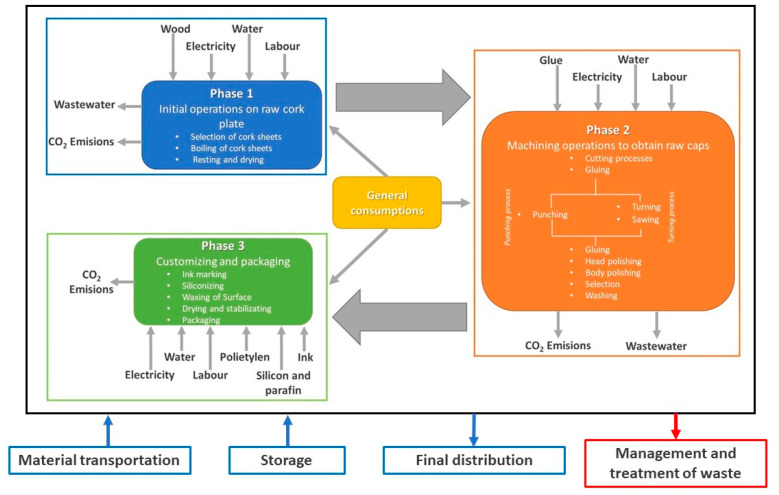
System limits The flowchart displays processes relations and consumption of material, energy, water, and other consumption flows and has been elaborated according to the reality of material, energy, water, and other consumption flows, and these depend on the existing production process [19,20].

**Figure 3 foods-10-00873-f003:**
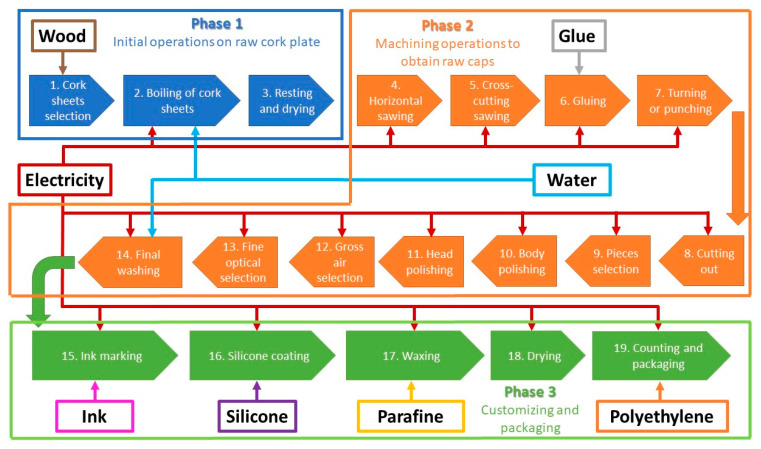
Flowchart.

**Table 1 foods-10-00873-t001:** Consumptions for 1000 kg raw material used to produce 2-piece stoppers with dimensions of 24 × 44—Drilled (Perforated) method.

	2-Piece Cork 24 × 44—Perforated		
	Phase 1	Phase 2	Phase 3
Raw material (kg)	99.6	798.5229723	−0.642434486
Water (kg)	2961.150685	234.818468	0
Energy (kWh)	394.0489893	339.7169957	4.757206185
Workforce (h)	16.47945205	54.0239726	3.034218613
Glue (kg)	0	6,1983	0
Silicone (kg)	0	0	0.106978801
Paraffin (kg)	0	0	0.535455685

**Table 2 foods-10-00873-t002:** Consumption for 1000 kg raw material used to produce 2-piece stoppers with dimensions 26 × 44—Drilled (Perforated) method.

	2-Piece Cork 26 × 44—Perforated		
	Phase 1	Phase 2	Phase 3
Raw material (kg)	173.5	734.8957104	−0.492139253
Water (kg)	2961.150685	179.9068018	0
Energy (kWh)	394.0070381	297.2638903	3.801867111
Workforce (h)	16.47945205	54.0239726	2.728139794
Glue (kg)	0	6.1983	0
Silicone (kg)	0	0	0.081962096
Paraffin (kg)	0	0	0.410177157

**Table 3 foods-10-00873-t003:** Consumptions for 1000 kg raw material used to produce 2-piece stoppers with dimensions of 24 × 44—Turning method.

	2-Piece Cork 24 × 44—Turned		
	Phase 1	Phase 2	Phase 3
Raw material (kg)	103.3	799.1195475	−4.939009633
Water (kg)	3281.575342	224.9152031	0
Energy (kWh)	419.7903808	397.2961628	4.556575146
Workforce (h)	15.21917808	23.1369863	0.228126727
Glue (kg)	0	6.1983	0
Silicone (kg)	0	0	0.102467063
Paraffin (kg)	0	0	0.512873306

**Table 4 foods-10-00873-t004:** Consumptions for 1000 kg raw material used to produce 2-piece stoppers with dimensions of 26 × 44—Turning method.

	2-Piece Cork 26 × 44—Turned		
	Phase 1	Phase 2	Phase 3
Raw material (kg)	157.9	743.6385754	−4.058037613
Water (kg)	3281.575342	193.3739137	0
Energy (kWh)	419.7607708	327.4015667	4.256729013
Workforce (h)	15.21917808	23.1369863	0.230178959
Glue (kg)	0	6.595	0
Silicone (kg)	0	0	0.088097455
Paraffin (kg)	0	0	0.440881397

**Table 5 foods-10-00873-t005:** Global impact for each geometry and production methodology, by impact categories.

		24 × 44 2P Perforated	26 × 44 2P Perforated	24 × 44 2P Turned	26 × 44 2P Turned
Abiotic depletion	kg Sb eq	0.002735461	0.002731983	0.003004334	0.00299546
Abiotic depletion (fossil fuels)	MJ	10690.65995	10545.34663	11749.76879	11510.02109
Global warming (GWP100a)	kg CO_2_ eq	977.7305994	963.2864969	1076.812504	1052.989073
Ozone layer depletion (ODP)	kg CFC-11 eq	0.000102725	0.000100916	0.00011299	0.000110023
Human toxicity	kg 1.4-DB eq	490.8382666	485.8388178	540.5864976	532.2016483
Fresh water aquatic ecotox.	kg 1.4-DB eq	526.3757405	521.1066613	580.335491	571.7369036
Marine aquatic ecotoxicity	kg 1.4-DB eq	1534271.083	1510568.815	1693274.685	1654770.983
Terrestrial ecotoxicity	kg 1.4-DB eq	8.938554382	8.890603038	9.803742104	9.725492703
Photochemical oxidation	kg C_2_H_4_ eq	0.37385754	0.369816267	0.404236078	0.397468484
Acidification	kg SO_2_ eq	5.719579397	5.605312106	6.306237612	6.119246576
Eutrophication	kg PO_4_—eq	2.247685829	2.225486034	2.476569152	2.440110841

**Table 6 foods-10-00873-t006:** Impact of stage 1 broken down by activities for the manufacturing process of 2-piece drilling-perforated stoppers, of 24 mm in diameter and 44 mm in length.

		Cork Sheets Selection	Boiling of Cork Sheets	Resting and Drying	Total
Abiotic depletion	kg Sb eq	7.659 × 10^−6^	0.00253652	8.9472 × 10^−6^	0.00255313
Abiotic depletion (fossil fuels)	MJ	210.954269	8703.63077	152.098393	9066.68343
Global warming (GWP100a)	kg CO_2_ eq	15.0859119	807.818226	14.7224994	837.626637
Ozone layer depletion (ODP)	kg CFC-11 eq	2.6202 × 10^−6^	8.0286 × 10^−5^	1.8379 × 10^−6^	8.4744 × 10^−5^
Human toxicity	kg 1.4-DB eq	2.49657001	428.496627	5.32512908	436.318326
Fresh water aquatic ecotox.	kg 1.4-DB eq	2.63458208	470.790845	5.22994656	478.655374
Marine aquatic ecotoxicity	kg 1.4-DB eq	2388.83562	1300467.37	23257.8208	1326114.03
Terrestrial ecotoxicity	kg 1.4-DB eq	0.50703198	7.95327691	0.04743803	8.50774692
Photochemical oxidation	kg C_2_H_4_ eq	0.07623872	0.25029771	0.00427678	0.33081321
Acidification	kg SO_2_ eq	0.08352229	4.48663646	0.1138804	4.68403916
Eutrophication	kg PO_4_—eq	0.02217068	1.99490029	0.0223638	2.03943478

**Table 7 foods-10-00873-t007:** Impact of stage 1 broken down by activities for the manufacturing process of 2-piece drilling-perforated stoppers, of 26 mm in diameter and 44 mm in length.

		Cork Sheets Selection	Boiling of Cork Sheets	Resting and Drying	Total
Abiotic depletion	kg Sb eq	7.659 × 10^−6^	0.00253651	8.9472 × 10^−6^	0.00255312
Abiotic depletion (fossil fuels)	MJ	210.954269	8703.47757	152.098393	9066.53024
Global warming (GWP100a)	kg CO_2_ eq	15.0859119	807.803397	14.7224994	837.611808
Ozone layer depletion (ODP)	kg CFC-11 eq	2.6202 × 10^−6^	8.0284 × 10^−5^	1.8379 × 10^−6^	8.4742 × 10^−5^
Human toxicity	kg 1.4-DB eq	2.49657001	428.491263	5.32512908	436.312962
Fresh water aquatic ecotox.	kg 1.4-DB eq	2.63458208	470.785577	5.22994656	478.650106
Marine aquatic ecotoxicity	kg 1.4-DB eq	2388.83562	1300443.95	23257.8208	1326090.6
Terrestrial ecotoxicity	kg 1.4-DB eq	0.50703198	7.95322913	0.04743803	8.50769914
Photochemical oxidation	kg C_2_H_4_ eq	0.07623872	0.2502934	0.00427678	0.3308089
Acidification	kg SO_2_ eq	0.08352229	4.48652176	0.1138804	4.68392445
Eutrophication	kg PO_4_—eq	0.02217068	1.99487777	0.0223638	2.03941225

**Table 8 foods-10-00873-t008:** Impact of stage 1 broken down by activities for the manufacturing process of 2-piece turned stoppers, of 24mm in diameter and 44mm in length.

		Cork Sheets Selection	Boiling of Cork Sheets	Resting and Drying	Total
Abiotic depletion	kg Sb eq	7.659 × 10^−6^	0.00279323	8.9472 × 10^−6^	0.00280984
Abiotic depletion (fossil fuels)	MJ	210.954269	9554.66708	152.098393	9917.71975
Global warming (GWP100a)	kg CO_2_ eq	15.0859119	886.657521	14.7224994	916.465933
Ozone layer depletion (ODP)	kg CFC-11 eq	2.6202 × 10^−6^	8.8029 × 10^−5^	1.8379 × 10^−6^	9.2488 × 10^−5^
Human toxicity	kg 1.4-DB eq	2.49657001	470.925973	5.32512908	478.747672
Fresh water aquatic ecotox.	kg 1.4-DB eq	2.63458208	517.560579	5.22994656	525.425108
Marine aquatic ecotoxicity	kg 1.4-DB eq	2388.83562	1427495.77	23257.8208	1453142.43
Terrestrial ecotoxicity	kg 1.4-DB eq	0.50703198	8.75342614	0.04743803	9.30789615
Photochemical oxidation	kg C_2_H_4_ eq	0.07623872	0.27479557	0.00427678	0.35531107
Acidification	kg SO_2_ eq	0.08352229	4.91662543	0.1138804	5.11402812
Eutrophication	kg PO_4_—eq	0.02217068	2.19302971	0.0223638	2.23756419

**Table 9 foods-10-00873-t009:** Impact of stage 1 broken down by activities for the manufacturing process of 2-piece turned stoppers, of 26 mm in diameter and 44 mm in length.

		Cork Sheets Selection	Boiling of Cork Sheets	Resting and Drying	Total
Abiotic depletion	kg Sb eq	7.659 × 10^−6^	0.00279322	8.9472 × 10^−6^	0.00280983
Abiotic depletion (fossil fuels)	MJ	210.954269	9554.55895	152.098393	9917.61162
Global warming (GWP100a)	kg CO_2_ eq	15.0859119	886.647055	14.7224994	916.455466
Ozone layer depletion (ODP)	kg CFC-11 eq	2.6202 × 10^−6^	8.8028 × 10^−5^	1.8379 × 10^−6^	9.2486 × 10^−5^
Human toxicity	kg 1.4-DB eq	2.49657001	470.922187	5.32512908	478.743887
Fresh water aquatic ecotox.	kg 1.4-DB eq	2.63458208	517.556861	5.22994656	525.42139
Marine aquatic ecotoxicity	kg 1.4-DB eq	2388.83562	1427479.24	23257.8208	1453125.9
Terrestrial ecotoxicity	kg 1.4-DB eq	0.50703198	8.75339242	0.04743803	9.30786243
Photochemical oxidation	kg C_2_H_4_ eq	0.07623872	0.27479253	0.00427678	0.35530803
Acidification	kg SO_2_ eq	0.08352229	4.91654447	0.1138804	5.11394716
Eutrophication	kg PO_4_—eq	0.02217068	2.19301381	0.0223638	2.23754829

**Table 10 foods-10-00873-t010:** Impact of stage 2 broken down by activities for the manufacturing process of two-piece perforated caps of 24 mm diameter and 44 mm in length.

d		Horizontal Sawing	Cross-Cutting Sawing	Gluing	Punching	Cutting Out	Pieces Selection
Abiotic depletion	kg Sb eq	9.79204 × 10^−7^	1.41377 × 10^−6^	0.000164796	1.09573 × 10^−6^	2.75878 × 10^−7^	1.58861 × 10^−6^
Abiotic depletion (fossil fuels)	MJ	16.64600967	24.03350876	1355.837713	18.62693063	4.689795459	27.00562884
Global warming (GWP100a)	kg CO_2_ eq	1.611265337	2.326344894	116.5679459	1.803010346	0.45395293	2.614033905
Ozone layer depletion (ODP)	kg CFC-11 eq	2.01144 × 10^−7^	2.90411 × 10^−7^	1.49206 × 10^−5^	2.2508 × 10^−7^	5.66696 × 10^−8^	3.26325 × 10^−7^
Human toxicity	kg 1.4-DB eq	0.582794785	0.841439111	45.93881528	0.652148968	0.164194806	0.945496246
Fresh water aquatic ecotox.	kg 1.4-DB eq	0.572377785	0.826399046	39.33888383	0.640492317	0.161259953	0.928596242
Marine aquatic ecotoxicity	kg 1.4-DB eq	2545.391196	3675.035809	170901.8665	2848.299754	717.1306705	4129.511593
Terrestrial ecotoxicity	kg 1.4-DB eq	0.005191731	0.007495821	0.350752412	0.005809561	0.001462702	0.008422797
Photochemical oxidation	kg C_2_H_4_ eq	0.000468061	0.000675786	0.036215966	0.000523761	0.00013187	0.000759358
Acidification	kg SO_2_ eq	0.012463342	0.017994573	0.855607705	0.013946514	0.003511383	0.020219884
Eutrophication	kg PO_4_—eq	0.002447548	0.003533769	0.172616166	0.002738812	0.000689565	0.003970775
		Body polishing	Head polishing	Gross air selection	Fine optical selection	Final washing	Total
Abiotic depletion	kg Sb eq	2.52933 × 10^−7^	3.23329 × 10^−7^	2.1573 × 10^−7^	1.95948 × 10^−7^	6.68326 × 10^−6^	0.00017782
Abiotic depletion (fossil fuels)	MJ	4.29973755	5.496440505	3.667317676	3.331027511	109.2488265	1572.882936
Global warming (GWP100a)	kg CO_2_ eq	0.416196927	0.532032856	0.354981282	0.322429776	10.57044332	137.5726375
Ozone layer depletion (ODP)	kg CFC-11 eq	5.19563 × 10^−8^	6.64167 × 10^−8^	4.43144 × 10^−8^	4.02508 × 10^−8^	1.31696 × 10^−6^	1.75401 × 10^−5^
Human toxicity	kg 1.4-DB eq	0.150538458	0.192436321	0.128396754	0.116622872	3.840612413	53.55349602
Fresh water aquatic ecotox.	kg 1.4-DB eq	0.147847701	0.188996673	0.126101763	0.11453833	3.778297956	46.82379159
Marine aquatic ecotoxicity	kg 1.4-DB eq	657.4857473	840.4771804	560.7805302	509.3573937	16701.70698	204087.0434
Terrestrial ecotoxicity	kg 1.4-DB eq	0.001341047	0.001714287	0.001143801	0.001038916	0.034737868	0.419110945
Photochemical oxidation	kg C_2_H_4_ eq	0.000120902	0.000154552	0.000103119	9.36635 × 10^−5^	0.003072618	0.042319658
Acidification	kg SO_2_ eq	0.003219336	0.004115342	0.002745825	0.002494035	0.081540327	1.017858266
Eutrophication	kg PO_4_—eq	0.000632212	0.00080817	0.000539224	0.000489778	0.016154085	0.204620104

**Table 11 foods-10-00873-t011:** Impact of stage 2 broken down by activities for the manufacturing process of 2-piece perforated stoppers with a diameter of 26 mm and 44 mm in length.

		Horizontal Sawing	Cross-Cutting Sawing	Gluing	Punching	Cutting Out	Pieces Selection
Abiotic depletion	kg Sb eq	7.98965 × 10^−7^	1.17416 × 10^−6^	0.000164991	9.77652 × 10^−7^	2.16819 × 10^−7^	1.24853 × 10^−6^
Abiotic depletion (fossil fuels)	MJ	13.58203958	19.96016537	1266.643583	16.61963209	3.685818907	21.22434939
Global warming (GWP100a)	kg CO_2_ eq	1.314685622	1.93206199	106.995319	1.608712096	0.356772125	2.054429809
Ozone layer depletion (ODP)	kg CFC-11 eq	1.6412 × 10^−7^	2.4119 × 10^−7^	1.37492 × 10^−7^	2.00825 × 10^−7^	4.45379 × 10^−8^	2.56466 × 10^−7^
Human toxicity	kg 1.4-DB eq	0.475521881	0.698826957	42.71807642	0.581871277	0.129044503	0.743087406
Fresh water aquatic ecotox.	kg 1.4-DB eq	0.4670223	0.686335973	35.80586103	0.571470784	0.126737933	0.729805301
Marine aquatic ecotoxicity	kg 1.4-DB eq	2076.870353	3052.16867	154931.7793	2541.357722	563.6096087	3245.478836
Terrestrial ecotoxicity	kg 1.4-DB eq	0.004236108	0.006225385	0.318317808	0.005183504	0.001149572	0.006619672
Photochemical oxidation	kg C_2_H_4_ eq	0.000381907	0.00056125	0.033585837	0.000467319	0.00010364	0.000596797
Acidification	kg SO_2_ eq	0.01016926	0.014944745	0.778614307	0.012443592	0.002759678	0.015891275
Eutrophication	kg PO_4_—eq	0.001997036	0.002934845	0.157790129	0.002443669	0.000541945	0.003120724
		Body polishing	Head polishing	Gross air selection	Fine optical selection	Final washing	Total
Abiotic depletion	kg Sb eq	1.98786 × 10^−7^	2.58946 × 10^−7^	1.72424 × 10^−7^	1.56613 × 10^−7^	5.1232 × 10^−6^	0.000175317
Abiotic depletion (fossil fuels)	MJ	3.379263359	4.401963622	2.9311332	2.662350576	83.74883637	1438.839135
Global warming (GWP100a)	kg CO_2_ eq	0.327098808	0.426091991	0.28372165	0.257704596	8.103176619	123.6597743
Ozone layer depletion (ODP)	kg CFC-11 eq	4.08336 × 10^−8^	5.31915 × 10^−8^	3.54186 × 10^−8^	3.21708 × 10^−8^	1.00957 × 10^−6^	1.58275 × 10^−5^
Human toxicity	kg 1.4-DB eq	0.118311662	0.154117503	0.10262214	0.09321177	2.944160451	48.75885197
Fresh water aquatic ecotox.	kg 1.4-DB eq	0.116196933	0.151362773	0.10078785	0.091545683	2.896388302	41.74351486
Marine aquatic ecotoxicity	kg 1.4-DB eq	516.7332817	673.1174421	448.2083569	407.1080007	12803.32936	181259.761
Terrestrial ecotoxicity	kg 1.4-DB eq	0.00105396	0.00137293	0.000914192	0.000830362	0.026629343	0.372532835
Photochemical oxidation	kg C_2_H_4_ eq	9.50198 × 10^−5^	0.000123777	8.24191 × 10^−5^	7.48613 × 10^−5^	0.002355432	0.038428258
Acidification	kg SO_2_ eq	0.002530151	0.003295876	0.002194623	0.001993378	0.062507946	0.90734483
Eutrophication	kg PO_4_—eq	0.00049687	0.000647243	0.000430979	0.000391459	0.012383488	0.183178388

**Table 12 foods-10-00873-t012:** Impact of stage 2 broken down by activities for the manufacturing process of 2-piece turned stoppers of 24 mm diameter and 44 mm in length.

		Horizontal Sawing	Cross-Cutting Sawing	Gluing	Turning	Cutting Out	Pieces Selection
Abiotic depletion	kg Sb eq	9.7518 × 10^−7^	2.00757 × 10^−6^	0.000176169	1.64611 × 10^−6^	3.02227 × 10^−7^	1.6872 × 10^−6^
Abiotic depletion (fossil fuels)	MJ	16.57760648	34.12776586	1549.171728	27.98319182	5.137714018	28.68165549
Global warming (GWP100a)	kg CO_2_ eq	1.604644189	3.303427504	135.2818839	2.708657984	0.497309606	2.776266397
Ozone layer depletion (ODP)	kg CFC-11 eq	2.00317 × 10^−7^	4.12386 × 10^−7^	1.72567 × 10^−5^	3.38137 × 10^−7^	6.2082 × 10^−8^	3.46577 × 10^−7^
Human toxicity	kg 1.4-DB eq	0.580399915	1.194849959	52.70764798	0.979721782	0.179876919	1.00417575
Fresh water aquatic ecotox.	kg 1.4-DB eq	0.570025722	1.173492952	45.98672891	0.962210023	0.17666176	0.986226895
Marine aquatic ecotoxicity	kg 1.4-DB eq	2534.931459	5218.578897	200465.1493	4278.993676	785.623239	4385.797847
Terrestrial ecotoxicity	kg 1.4-DB eq	0.005170397	0.010644124	0.411051441	0.00872769	0.001602404	0.008945534
Photochemical oxidation	kg C_2_H_4_ eq	0.000466137	0.000959622	0.041652228	0.000786845	0.000144465	0.000806485
Acidification	kg SO_2_ eq	0.012412126	0.025552431	1.000362389	0.020951813	0.003846753	0.021474773
Eutrophication	kg PO_4_—eq	0.00243749	0.005017979	0.20104305	0.004114511	0.000755424	0.00421721
		Body polishing	Head polishing	Gross air selection	Fine optical selection	Final washing	Total
Abiotic depletion	kg Sb eq	2.6863 × 10^−7^	3.22979 × 10^−7^	2.08448 × 10^−7^	1.87684 × 10^−7^	6.4014 × 10^−6^	0.000190176
Abiotic depletion (fossil fuels)	MJ	4.566588388	5.490500551	3.543511261	3.190544319	104.6413522	1783.112158
Global warming (GWP100a)	kg CO_2_ eq	0.442026992	0.531457893	0.34299733	0.30883158	10.12464405	157.9221475
Ozone layer depletion (ODP)	kg CFC-11 eq	5.51808 × 10^−8^	6.6345 × 10^−8^	4.28183 × 10^−8^	3.85532 × 10^−8^	1.26142 × 10^−6^	2.00806 × 10^−5^
Human toxicity	kg 1.4-DB eq	0.159881194	0.192228357	0.124062158	0.111704404	3.678637921	60.91318634
Fresh water aquatic ecotox.	kg 1.4-DB eq	0.157023443	0.188792426	0.121844644	0.109707775	3.618951522	54.05166607
Marine aquatic ecotoxicity	kg 1.4-DB eq	698.2907082	839.5688843	541.8489203	487.875688	15997.32699	236233.9856
Terrestrial ecotoxicity	kg 1.4-DB eq	0.001424275	0.001712435	0.001105187	0.0009951	0.033272829	0.484651415
Photochemical oxidation	kg C_2_H_4_ eq	0.000128406	0.000154385	9.96382 × 10^−5^	8.97133 × 10^−5^	0.002943033	0.048230958
Acidification	kg SO_2_ eq	0.003419135	0.004110894	0.002653128	0.002388851	0.078101434	1.175273728
Eutrophication	kg PO_4_—eq	0.000671449	0.000807296	0.000521021	0.000469122	0.015472801	0.235527353

**Table 13 foods-10-00873-t013:** Impact of stage 2 broken down by activities for the manufacturing process of 2-piece turned stoppers of 26 mm diameter and 44 mm in length.

		Horizontal Sawing	Cross-Cutting Sawing	Gluing	Turning	Cutting Out	Pieces Selection
Abiotic depletion	kg Sb eq	8.47964 × 10^−7^	1.64618 × 10^−6^	0.000169818	1.34947 × 10^−6^	2.63956 × 10^−7^	1.48859 × 10^−6^
Abiotic depletion (fossil fuels)	MJ	14.41499739	27.98431493	1348.693492	22.94035787	4.487133999	25.30529903
Global warming (GWP100a)	kg CO_2_ eq	1.395312516	2.708766697	114.9374131	2.220532379	0.434336133	2.449448966
Ozone layer depletion (ODP)	kg CFC-11 eq	1.74185 × 10^−7^	3.38151 × 10^−7^	1.47406 × 10^−5^	2.77202 × 10^−7^	5.42207 × 10^−8^	3.05779 × 10^−7^
Human toxicity	kg 1.4-DB eq	0.504684634	0.979761104	45.59073245	0.803166717	0.15709941	0.885965862
Fresh water aquatic ecotox.	kg 1.4-DB eq	0.495663792	0.962248642	38.62717059	0.788810741	0.154291381	0.870129917
Marine aquatic ecotoxicity	kg 1.4-DB eq	2204.240427	4279.165415	167478.2768	3507.878833	686.1410996	3869.509068
Terrestrial ecotoxicity	kg 1.4-DB eq	0.0044959	0.00872804	0.343908389	0.007154878	0.001399494	0.007892481
Photochemical oxidation	kg C_2_H_4_ eq	0.000405328	0.000786877	0.035892957	0.000645048	0.000126171	0.000711547
Acidification	kg SO_2_ eq	0.010792919	0.020952654	0.840047413	0.0171761	0.003359645	0.018946798
Eutrophication	kg PO_4_—eq	0.00211951	0.004114676	0.169854345	0.003373038	0.000659766	0.003720767
		Body polishing	Head polishing	Gross air selection	Fine optical selection	Final washing	Total
Abiotic depletion	kg Sb eq	2.37007 × 10^−7^	2.83878 × 10^−7^	1.89408 × 10^−7^	1.75351 × 10^−7^	5.50972 × 10^−6^	0.00018181
Abiotic depletion (fossil fuels)	MJ	4.029017249	4.825795701	3.219852168	2.980878765	90.06936077	1548.9505
Global warming (GWP100a)	kg CO_2_ eq	0.389992314	0.467117194	0.311668459	0.288536816	8.714725399	134.31785
Ozone layer depletion (ODP)	kg CFC-11 eq	4.8685 × 10^−8^	5.83129 × 10^−8^	3.89074 × 10^−8^	3.60197 × 10^−8^	1.08576 × 10^−6^	1.71579 × 10^−5^
Human toxicity	kg 1.4-DB eq	0.141060247	0.168956322	0.112730504	0.104363786	3.166348848	52.61486989
Fresh water aquatic ecotox.	kg 1.4-DB eq	0.138538906	0.16593636	0.110715534	0.102498366	3.114968489	45.53097272
Marine aquatic ecotoxicity	kg 1.4-DB eq	616.0890952	737.9268748	492.357239	455.8151002	13769.5984	198096.9984
Terrestrial ecotoxicity	kg 1.4-DB eq	0.001256612	0.00150512	0.001004241	0.000929708	0.028638747	0.406913609
Photochemical oxidation	kg C_2_H_4_ eq	0.00011329	0.000135694	9.05374 × 10^−5^	8.38178 × 10^−5^	0.002533196	0.041524464
Acidification	kg SO_2_ eq	0.00301664	0.003613211	0.002410795	0.002231869	0.067225541	0.989773585
Eutrophication	kg PO_4_—eq	0.000592407	0.000709561	0.000473431	0.000438294	0.013318027	0.199373823

**Table 14 foods-10-00873-t014:** Impact of stage 3 broken down by activities for the manufacturing process of perforated stoppers of 2-pieces of 24 mm in diameter and 44 mm in length.

		Ink Marking	Silicone Coating	Waxing	Drying	Counting and Packaging	Total
Abiotic depletion	kg Sb eq	3.99196 × 10^−7^	1.0427 × 10^−6^	2.83187 × 10^−6^	9.31642 × 10^−8^	1.45415 × 10^−7^	4.51234 × 10^−6^
Abiotic depletion (fossil fuels)	MJ	3.879728885	8.746627243	34.41152473	1.583747734	2.471987204	51.09361579
Global warming (GWP100a)	kg CO_2_ eq	0.353097735	0.718555749	1.067095298	0.153300273	0.239278204	2.531327259
Ozone layer depletion (ODP)	kg CFC-11 eq	4.15944 × 10^−8^	2.53403 × 10^−7^	9.6526 × 10^−8^	1.91374 × 10^−8^	2.98705 × 10^−8^	4.40532 × 10^−7^
Human toxicity	kg 1.4-DB eq	0.124486862	0.267216194	0.432748548	0.05544872	0.086546943	0.966447267
Fresh water aquatic ecotox.	kg 1.4-DB eq	0.122416704	0.262765725	0.371938444	0.054457617	0.084999984	0.896578474
Marine aquatic ecotoxicity	kg 1.4-DB eq	476.0119299	1337.515368	1636.314337	242.1756096	377.9989673	4070.016211
Terrestrial ecotoxicity	kg 1.4-DB eq	0.006125014	0.001765573	0.002541001	0.000493956	0.000770989	0.011696533
Photochemical oxidation	kg C_2_H_4_ eq	0.000110929	0.000201278	0.000298421	4.45326 × 10^−5^	6.95086 × 10^−5^	0.00072467
Acidification	kg SO_2_ eq	0.002352501	0.004387312	0.007905528	0.001185797	0.001850847	0.017681985
Eutrophication	kg PO_4_—eq	0.000531177	0.001008274	0.001495183	0.000232866	0.000363469	0.00363097

**Table 15 foods-10-00873-t015:** Impact of stage 3 broken down by activities for the manufacturing process of perforated stoppers of 2-pieces of 26 mm in diameter and 44 mm in length.

		Ink Marking	Silicone Coating	Waxing	Drying	Counting and Packaging	Total
Abiotic depletion	kg Sb eq	3.66576 × 10^−7^	8.06203 × 10^−7^	2.18252 × 10^−6^	7.44507 × 10^−8^	1.16206 × 10^−7^	3.54596 × 10^−6^
Abiotic depletion (fossil fuels)	MJ	3.32520477	6.825955703	26.58505866	1.265626713	1.975449103	39.97729495
Global warming (GWP100a)	kg CO_2_ eq	0.29942208	0.562594226	0.839177953	0.122507465	0.191215356	2.014917081
Ozone layer depletion (ODP)	kg CFC-11 eq	3.48937 × 10^−8^	1.95653 × 10^−7^	7.6657 × 10^−8^	1.52933 × 10^−8^	2.38705 × 10^−8^	3.46367 × 10^−8^
Human toxicity	kg 1.4-DB eq	0.105072373	0.209094418	0.339365866	0.044310959	0.069162608	0.767006223
Fresh water aquatic ecotox.	kg 1.4-DB eq	0.103349234	0.20560664	0.292642674	0.043518935	0.06792638	0.713043863
Marine aquatic ecotoxicity	kg 1.4-DB eq	391.2179863	1043.809816	1287.826663	193.5307715	302.0718391	3218.457075
Terrestrial ecotoxicity	kg 1.4-DB eq	0.005952063	0.001391593	0.002016564	0.000394737	0.000616124	0.01037108
Photochemical oxidation	kg C_2_H_4_ eq	9.53368 × 10^−5^	0.000157716	0.000234918	3.55875 × 10^−5^	5.55467 × 10^−5^	0.000579106
Acidification	kg SO_2_ eq	0.001937313	0.00345472	0.006224117	0.000947611	0.001479075	0.014042836
Eutrophication	kg PO_4_—eq	0.000449642	0.000790828	0.001178395	0.000186092	0.00029046	0.002895417

**Table 16 foods-10-00873-t016:** Impact of stage 3 broken down by activities for the manufacturing process of 2-piece turning stoppers of 24 mm in diameter and 44 mm in length.

		Ink Marking	Silicone Coating	Waxing	Drying	Counting and Packaging	Total
Abiotic depletion	kg Sb eq	3.8197 × 10^−7^	9.98725 × 10^−6^	2.71244 × 10^−6^	8.9235 × 10^−8^	1.39282 × 10^−7^	4.32165 × 10^−6^
Abiotic depletion (fossil fuels)	MJ	3.714262026	8.377745836	32.96024856	1.516954549	2.367733291	48.93694427
Global warming (GWP100a)	kg CO_2_ eq	0.338064796	0.688251284	1.02209148	0.146834967	0.229186853	2.424429382
Ozone layer depletion (ODP)	kg CFC-11 eq	3.98266 × 10^−8^	2.42716 × 10^−7^	9.24551 × 10^−8^	1.83303 × 10^−8^	2.86107 × 10^−8^	4.21939 × 10^−7^
Human toxicity	kg 1.4-DB eq	0.119190935	0.255946583	0.414497754	0.053110218	0.082896901	0.925642391
Fresh water aquatic ecotox.	kg 1.4-DB eq	0.117208647	0.251683809	0.356252263	0.052160914	0.081415184	0.858720818
Marine aquatic ecotoxicity	kg 1.4-DB eq	455.8481318	1281.106819	1567.304201	231.9620636	362.0571892	3898.278405
Terrestrial ecotoxicity	kg 1.4-DB eq	0.005858016	0.001691112	0.002433836	0.000473124	0.000738473	0.011194562
Photochemical oxidation	kg C_2_H_4_ eq	0.000106196	0.00019279	0.000285836	4.26545 × 10^−5^	6.65771 × 10^−5^	0.000694053
Acidification	kg SO_2_ eq	0.002252817	0.004202281	0.007572119	0.001135787	0.001772789	0.016935794
Eutrophication	kg PO_4_—eq	0.000508569	0.000965751	0.001432125	0.000223046	0.00034814	0.00347763

**Table 17 foods-10-00873-t017:** Impact of stage 3 broken down by activities for the manufacturing process of turning stoppers of 2-pieces of 26 mm in diameter and 44 mm in length.

		Ink Marking	Silicone Coating	Waxing	Drying	Counting and Packaging	Total
Abiotic depletion	kg Sb eq	3.71742 × 10^−7^	8.74484 × 10^−7^	2.36019 × 10^−6^	8.33581 × 10^−8^	1.30109 × 10^−7^	3.81988 × 10^−6^
Abiotic depletion (fossil fuels)	MJ	3.540389412	7.471756116	28.81803365	1.417048464	2.211795222	43.45902287
Global warming (GWP100a)	kg CO_2_ eq	0.321234641	0.617759411	0.925509406	0.137164469	0.214092689	2.215760616
Ozone layer depletion (ODP)	kg CFC-11 eq	3.77256 × 10^−8^	2.11928 × 10^−7^	8.53306 × 10^−8^	1.7123 × 10^−8^	2.67264 × 10^−8^	3.78833 × 10^−7^
Human toxicity	kg 1.4-DB eq	0.113103466	0.229467166	0.373274436	0.049612398	0.077437341	0.842894806
Fresh water aquatic ecotox.	kg 1.4-DB eq	0.111229987	0.225633927	0.322901711	0.048725615	0.076053209	0.784544449
Marine aquatic ecotoxicity	kg 1.4-DB eq	429.2607484	1142.563523	1421.374297	216.685125	338.2122321	3548.095925
Terrestrial ecotoxicity	kg 1.4-DB eq	0.005803787	0.001537816	0.002243281	0.000441964	0.000689838	0.010716686
Photochemical oxidation	kg C_2_H_4_ eq	0.000101307	0.000173314	0.000259334	3.98453 × 10^−5^	6.21924 × 10^−5^	0.000635992
Acidification	kg SO_2_ eq	0.002122634	0.003814283	0.006871913	0.001060985	0.001656034	0.015525848
Eutrophication	kg PO_4_—eq	0.000483003	0.000869852	0.001302323	0.000208356	0.000325212	0.003188745

**Table 18 foods-10-00873-t018:** Impacts generated by each stage of the activity of manufacturing 2-piece stoppers with dimensions of 24 × 44 by drilling.

Impact Category	Unit	Stage 1	Stage 2	Stage 3	Total
Abiotic depletion	kg Sb eq	0.0026	0.0002	5.00 × 10^−6^	0.002805
Abiotic depletion (fossil fuels)	MJ	9066.7	1572.9	51.094	10690.694
Global warming (GWP100a)	kg CO_2_ eq	837.63	137.57	2.5313	977.7313
Ozone layer depletion (ODP)	kg CFC-11 eq	8.00 × 10^−6^	2.00 × 10^−5^	4.00 × 10^−7^	0.0001004
Human toxicity	kg 1.4-DB eq	436.32	53.553	0.9664	490.8394
Fresh water aquatic ecotox.	kg 1.4-DB eq	478.66	46.824	0.8966	526.3806
Marine aquatic ecotoxicity	kg 1.4-DB eq	1.00 × 10^6^	204087	4070	1208157
Terrestrial ecotoxicity	kg 1.4-DB eq	8.5077	0.4191	0.0117	8.9385
Photochemical oxidation	kg C_2_H_4_ eq	0.3308	0.0423	0.0007	0.3738
Acidification	kg SO_2_ eq	4.684	1.0179	0.0177	5.7196
Eutrophication	kg PO_4_—eq	2.0394	0.2046	0.0036	2.2476

**Table 19 foods-10-00873-t019:** Impacts generated by each stage of the 2-piece stopper manufacturing activity with dimensions of 26 × 44 by drilling.

Impact Category	Unit	Stage 1	Stage 2	Stage 3	Total
Abiotic depletion	kg Sb eq	0.0027	0.0026	0.0002	0.0055
Abiotic depletion (fossil fuels)	MJ	10545	9066.5	1438.8	21050.3
Global warming (GWP100a)	kg CO_2_ eq	963.29	837.61	123.66	1924.56
Ozone layer depletion (ODP)	kg CFC-11 eq	0.0001	8.00 × 10^−5^	2.00 × 10^−5^	0.0002
Human toxicity	kg 1.4-DB eq	485.84	436.31	48.759	970.909
Fresh water aquatic ecotox.	kg 1.4-DB eq	521.11	478.65	41.744	1041.504
Marine aquatic ecotoxicity	kg 1.4-DB eq	2.00 × 10^6^	1.00 × 10^6^	181260	3181260
Terrestrial ecotoxicity	kg 1.4-DB eq	8.8906	8.5077	0.3725	17.7708
Photochemical oxidation	kg C_2_H_4_ eq	0.3698	0.3308	0.0384	0.739
Acidification	kg SO_2_ eq	5.6053	4.6839	0.9073	11.1965
Eutrophication	kg PO_4_—eq	2.2255	2.0394	0.1832	4.4481

**Table 20 foods-10-00873-t020:** Impacts generated by each stage of the activity of manufacturing 2-piece stopper with dimensions of 24 × 44 by turning.

Impact Category	Unit	Stage 1	Stage 2	Stage 3	Total
Abiotic depletion	kg Sb eq	0.003	0.0028	0.0002	0.006
Abiotic depletion (fossil fuels)	MJ	11750	9917.7	1783.1	23450.8
Global warming (GWP100a)	kg CO_2_ eq	1076.8	916.47	157.92	2151.19
Ozone layer depletion (ODP)	kg CFC-11 eq	0.0001	9.00 × 10^−5^	2.00 × 10^−5^	0.00021
Human toxicity	kg 1.4-DB eq	540.59	478.75	60.913	1080.253
Fresh water aquatic ecotox.	kg 1.4-DB eq	580.34	525.43	54.052	1159.822
Marine aquatic ecotoxicity	kg 1.4-DB eq	2.00 × 10^6^	1.00 × 10^6^	236234	3236234
Terrestrial ecotoxicity	kg 1.4-DB eq	9.8037	9.3079	0.4847	19.5963
Photochemical oxidation	kg C_2_H_4_ eq	0.4042	0.3553	0.0482	0.8077
Acidification	kg SO_2_ eq	6.3062	5.114	1.1753	12.5955
Eutrophication	kg PO_4_—eq	2.4766	2.2376	0.2355	4.9497

**Table 21 foods-10-00873-t021:** Impacts generated by each stage of the 2-piece stopper manufacturing activity with dimensions of 26 × 44 by turning.

Impact Category	Unit	Stage 1	Stage 2	Stage 3	Total
Abiotic depletion	kg Sb eq	0.003	0.0028	0.0002	0.006
Abiotic depletion (fossil fuels)	MJ	11510	9917.6	1549	22976.6
Global warming (GWP100a)	kg CO_2_ eq	1053	916.46	134.32	2103.78
Ozone layer depletion (ODP)	kg CFC-11 eq	0.0001	9.00 × 10^−5^	2.00 × 10^−5^	0.00021
Human toxicity	kg 1.4-DB eq	532.2	478.74	52.615	1063.555
Fresh water aquatic ecotox.	kg 1.4-DB eq	571.74	525.42	45.531	1142.691
Marine aquatic ecotoxicity	kg 1.4-DB eq	2.00 × 10^6^	1.00 × 10^6^	198097	3198097
Terrestrial ecotoxicity	kg 1.4-DB eq	9.7255	9.3079	0.4069	19.4403
Photochemical oxidation	kg C_2_H_4_ eq	0.3975	0.3553	0.0415	0.7943
Acidification	kg SO_2_ eq	6.1192	5.1139	0.9898	12.2229
Eutrophication	kg PO_4_—eq	2.4401	2.2375	0.1994	4.877

**Table 22 foods-10-00873-t022:** Global impact of each geometry and production methodology, by impact category and for one cork stopper unit.

		24 × 44 2P Perforated	26 × 44 2P Perforated	24 × 44 2P Turned	26 × 44 2P Turned
Abiotic depletion	kg Sb eq	1.042 × 10^−5^	1.22135 × 10^−5^	1.14442 × 10^−5^	1.33914 × 10^−5^
Abiotic depletion (fossil fuels)	MJ	40.72334221	47.14373321	44.75774719	51.45638001
Global warming (GWP100a)	kg CO_2_ eq	3.724415328	4.306441809	4.101842571	4.70746365
Ozone layer depletion (ODP)	kg CFC-11 eq	3.91305 × 10^−7^	4.51152 × 10^−7^	4.30407 × 10^−7^	4.91866 × 10^−7^
Human toxicity	kg 1.4-DB eq	1.869723178	2.1719775	2.059226375	2.37924588
Fresh water aquatic ecotox.	kg 1.4-DB eq	2.005094121	2.32964494	2.210640027	2.555991092
Marine aquatic ecotoxicity	kg 1.4-DB eq	5844.414344	6753.106912	6450.098008	7397.773112
Terrestrial ecotoxicity	kg 1.4-DB eq	0.034049143	0.039746083	0.037344855	0.043478517
Photochemical oxidation	kg C_2_H_4_ eq	0.001424115	0.00165329	0.001539834	0.001776912
Acidification	kg SO_2_ eq	0.021787279	0.025058952	0.024022004	0.027356533
Eutrophication	kg PO_4_—eq	0.008561986	0.009949196	0.009433858	0.010908692

## Data Availability

The data presented in this study are available on request from the corresponding author. The data are not publicly available due to privacy.

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
