# Peer review of "2-Piece Cork Stoppers as Alternative for Valorization of Thin Cork Planks: Analysis by LCA Methodology"

_foods, 2021, doi:10.3390/foods10040873_

Round 1

Reviewer 1 Report

This paper is focused on the evaluation of the environmental impact related to the production of 2-piece cork stoppers with two different technologies, drilling and turning. This study is very interesting also because it describes alternative ways of producing cork stoppers that can exploit thinner sheets, given the limited availability of cork; anyway, I would like some clarifications and suggest some reviews to the work.

Major reviews:

- Line 185: in my opinion, the choice of the functional unit (1kg) does not reflect the function of the product that is the sealing of one (or more) bottle. In fact, in this case the mass is not directly related to the function; 1kg of 24 mm caps performs the function of closing a certain number of bottles which is certainly greater than that provided by 1kg of 26 mm caps (since in 1kg of 26 mm caps there are fewer caps). Therefore, it seems to me that the function performed by the two different kg is not the same and the comparison does not take place for the same function; moreover, this choice subsequently influences the results by making them more difficult to understand.
I believe it is more correct that the functional unit refers directly to the product, so I suggest to modify the F.U. considering for example one cork stopper.
Otherwise, if you want to consider 1kg as F.U., I think it is necessary to specify the number of caps per kg for each size (although I think the first proposal is more correct).

- Line 242, Table 1: It is not clear to me what quantity of caps the data in the table refer to, 1kg? Please provide this information in the captions for Table 1, 2, 3 and 4.

- Line 242, Table 1: It’s not clear to me what the "Raw material" input in Phase 1 refers to; does it represent the amount of cork needed to obtain 1kg of corks?
It’s also not clear to me what the “Raw material” input in Phase 2 represents; does it represent the amount of cork after boiling and drying (since it seems from the figure that no other raw materials are added in Phase 2)?
Finally, it’s not clear to me what the negative “Raw material” input in Phase 2 refers to; it represents the waste of cork?
Thus, I suggest to provide more information on the inventory data in Tables 1, 2, 3 and 4.

- Line 242, Table 1: The units of measurement of some inventory data are missing, please add them.

- Line 340, Table 1: as above, please specify in the captions to which quantity of caps the results of Tables 1, 2, 3 and 4 refer.

Minor reviews:

- Line 113: please provide the definition of TCA before using the abbreviation.

- Line 198, Figure 2: please improve the quality of the image, in particular the writing is very small.

- Line 216: I think that the management and treatment of production waste is of fundamental importance within the LCA. Why this exclusion?

- Line 311, general comment: have you considered the risk of paraffin coming into contact with wine?

- Line 356, Table 6: please check the “Total” values, it seems to me that they do not represent the sum of the partial values.

- Line 356, Table 6: It seems to me that the impacts of Phase 1 are the same for each type of cap and technology, as stated in Lines 362-364. However, Tables 15, 16, 17 and 18 show different environmental results for each Stage 1. Therefore Table 6 refers to a specific type of caps? Moreover, it might be useful to explain better what this difference is due to (see also Lines 497-507).

- Lines 378, 382 (Tables 7 and 8): I think it should be “perforated caps” instead of “turned”.

- Line 398: I think it should be “In the case of turned caps” instead of “perforated”.

- Line 430: I think it should be “Analysis of the impacts generated in stage 3” instead of “stage 2”.

- Line 432: I think it should be “..is made up of 5 different processes” instead of “3 different processes”.

- Line 452: It is not clear to me why there are differences between caps of the same size but different technology in Phase 3. That is, how does the production technology (drilling or turning) affect Stage 3?

- Tables 16, 17, 18: please check the “Total” values, it seems to me that they do not represent the sum of the partial values.

- General comment: in my opinion, it should be useful to indicate the amount of cork scrap for each type of cap and technology, originated from the cork platform, and analyse if and how much it affects the results.
In addition, if I understand correctly, this cork scrap has been treated as by-product; what type of allocation has been used (economic?) and why?

Author Response

Dear reviewer.

We would like to thank again the work performed of the reviewers and your suggestions. We have worked hard in order to improve the paper based on the comments received.

Below we explain such last corrections, point to point. First, we show the text of the reviews, while our answers appear next in red colour, in order to facilitate the checking of the corrections and the improvements in accordance to every suggestion.

Major reviews:

- Line 185: in my opinion, the choice of the functional unit (1kg) does not reflect the function of the product that is the sealing of one (or more) bottle. In fact, in this case the mass is not directly related to the function; 1kg of 24 mm caps performs the function of closing a certain number of bottles which is certainly greater than that provided by 1kg of 26 mm caps (since in 1kg of 26 mm caps there are fewer caps). Therefore, it seems to me that the function performed by the two different kg is not the same and the comparison does not take place for the same function; moreover, this choice subsequently influences the results by making them more difficult to understand.
I believe it is more correct that the functional unit refers directly to the product, so I suggest to modify the F.U. considering for example one cork stopper.
Otherwise, if you want to consider 1kg as F.U., I think it is necessary to specify the number of caps per kg for each size (although I think the first proposal is more correct).

Initially, during the research and drafting of the article, the use of a cork stopper as a functional unit was considered. However, the use of a cork stopper as a functional unit was discarded because the impacts of a 24 mm and a 26 mm cork stopper are substantially affected by the difference in volume of the two stoppers.

Although the LCA information for a cork stopper is very interesting for a winery, this information provides very little information to the rest of the actors in the cork stopper production chain.

On the other hand, as the reviewer cites, the data that wineries are really interested in is the impact associated with a cork stopper regardless of its geometry. However, they are not interested in the distribution of impacts for each of the phases.

In any case, we understand and share the reviewer's proposal. That is why in the results chapter we have added table 19 which shows the overall impacts associated with each of the corks analysed so that the user of these corks can know, specifically, the impacts associated with the use of one of these corks for each of the geometries studied.

- Line 242, Table 1: It is not clear to me what quantity of caps the data in the table refer to, 1kg? Please provide this information in the captions for Table 1, 2, 3 and 4.

Thank you very much for this comment. As you request, we have changed the footer of each of the tables to include the mass of cork stoppers to which they refer.

- Line 242, Table 1: It’s not clear to me what the "Raw material" input in Phase 1 refers to; does it represent the amount of cork needed to obtain 1kg of corks?
It’s also not clear to me what the “Raw material” input in Phase 2 represents; does it represent the amount of cork after boiling and drying (since it seems from the figure that no other raw materials are added in Phase 2)?
Finally, it’s not clear to me what the negative “Raw material” input in Phase 2 refers to; it represents the waste of cork?
Thus, I suggest to provide more information on the inventory data in Tables 1, 2, 3 and 4.

Thank you very much for this comment. Phase 2 refers to the processes by which the raw cork planks are transformed into cylindrical stoppers.

In this process, a large quantity of cork is consumed (as can be seen from the data provided, almost 80% of the total mass of cork from the selected planks is not usable for the production of stoppers).

Note that tables 1 to 4 show the consumption in each phase, so that, although there is no actual input of materials, there is consumption associated with the waste resulting from the processing of the plank.

- Line 242, Table 1: The units of measurement of some inventory data are missing, please add them.

Thank you very much for this comment. As you request, we have included all units in table 1.

- Line 340, Table 1: as above, please specify in the captions to which quantity of caps the results of Tables 1, 2, 3 and 4 refer.

 Thank you very much for this comment. As you request, we have included all units in tables 1 to 4.

Minor reviews:

- Line 113: please provide the definition of TCA before using the abbreviation.

Thank you very much for this comment. As you request, we have described the term ATT including its full name and a brief explanation of the term.

- Line 198, Figure 2: please improve the quality of the image, in particular the writing is very small.

Thank you very much for this comment. As you request, we have increased the total size of the image including the size of the texts included in the figure itself.

- Line 216: I think that the management and treatment of production waste is of fundamental importance within the LCA. Why this exclusion?

Thank you very much for this comment.

The cork stopper production process, as can be seen, involves the use of a very small part of the total volume of raw cork for the manufacture of stoppers.

However, the waste generated is recovered as raw material for other processes. For example, the fragments of "clean" cork from the drilling or turning phases are used to obtain cork washers or granulated material for the manufacture of fragmented, micro-granulated or granulated cork stoppers. On the other hand, the dust from the polishing processes is used for the colmatado processes of lower quality cork stoppers and the waste from other cutting and processing processes is used for the manufacture of other elements, reaching net utilisation rates of over 95% of the raw material used in the process. This is fundamental for the economic sustainability of cork stopper manufacturers, given that the cost of the raw material is one of the main costs they bear.

Based on the above and since all wastes are considered within the study, we have decided not to consider reuse or waste treatment processes within our system boundaries.

- Line 311, general comment: have you considered the risk of paraffin coming into contact with wine?

Thank you very much for this comment.

The paraffin used in the process is food-grade and does not pose a health risk.

Attending to organoleptic consideration, these paraffins do not contribute any flavour or odour.

Anyway, it should be noted that the paraffin waxes applied to the corks are necessary to prevent the wine from penetrating the cork and, in particular, to prevent surface capillarity.

Line 356, Table 6: please check the “Total” values, it seems to me that they do not represent the sum of the partial values.

Thank you very much for this comment. As you request,

Line 356, Table 6: It seems to me that the impacts of Phase 1 are the same for each type of cap and technology, as stated in Lines 362-364. However, Tables 15, 16, 17 and 18 show different environmental results for each Stage 1. Therefore Table 6 refers to a specific type of caps? Moreover, it might be useful to explain better what this difference is due to (see also Lines 497-507).

Thank you very much for this comment.

As you indicate, there was some confusion. Initially, we considered that this process should yield undifferentiated values for all processes, as it seems logical to think, but when analysing the primary data obtained, small differences were observed, which were finally considered for this phase.

We have included tables 6, 7, 8 and 9 with the phase 1 data for each of the processes.

We have also included an explanatory paragraph on this point.

- Lines 378, 382 (Tables 7 and 8): I think it should be “perforated caps” instead of “turned”.

Thank you very much for this comment. As you request, we have corrected the sum of the indicated values.

Line 398: I think it should be “In the case of turned caps” instead of “perforated”.

Thank you very much for this comment. As you request, we have corrected the reference to the cork stopper manufacturing methodology

Line 430: I think it should be “Analysis of the impacts generated in stage 3” instead of “stage 2”.

Thank you very much for this comment. As you request, we have corrected the stage number.

Line 432: I think it should be “..is made up of 5 different processes” instead of “3 different processes”.

Thank you very much for this comment. As you request, we have corrected the number of processes.

Line 452: It is not clear to me why there are differences between caps of the same size but different technology in Phase 3. That is, how does the production technology (drilling or turning) affect Stage 3?

Thank you very much for this comment.

There is an important difference in consumption between 24 and 26 mm corks, given that their surface/mass ratio is different in 24 and 26 mm corks, and this has a significant effect on the needs for silicone, paraffin, ink, etc.

This was one of the reasons for using 1 kg of finished corks as a functional unit and not one cork.

On the other hand, for each of the processes there are rejected stoppers and although the surface quality obtained in both processes is similar, it is not the same, which implies slight differences in surface porosity and consequently also small differences in the consumption of ink, paraffin and silicone.

Tables 16, 17, 18: please check the “Total” values, it seems to me that they do not represent the sum of the partial values.

Thank you very much for this comment. As you request, we have corrected the summations in the tables.

General comment: in my opinion, it should be useful to indicate the amount of cork scrap for each type of cap and technology, originated from the cork platform, and analyse if and how much it affects the results.
In addition, if I understand correctly, this cork scrap has been treated as by-product; what type of allocation has been used (economic?) and why?

Thank you very much for this comment. In the consumption section, the cork waste generated for each of the phases and for each of the processes has been indicated.

However, it is not possible to associate this waste to a specific type of valuation, given that each producer will use these resources in a different way depending on factors such as their location, the presence of nearby industries or potential commercial agreements for this waste.

For this reason, we wish to record the different raw material wastes produced at each stage of the process to allow each manufacturer to estimate the best way to make use of them.

Special thanks again for your good comments, which help us to improve our article and make it a document of great scientific interest.

Reviewer 2 Report

Dear Authors,

the paper covers an interesting issue for the journal and for the readers. Hovwever, I believe it should be improved.

Even if I appreciate the method adopted for the analysis, more explanations are required to improve the readability of the paper.

As first, you should better explain the method as for the theory to be adopted. Then, in the results' section, all the numbers and data provided in  the tables have to be explained, at the least the first time a data appears. 

Similarly, all the assumptions, limits, variables should be justified (previous research?), as well as the flowchart.

Finally, I'd suggest to clearly state the objective of the study at the beginning, and then summarize if and how the objective has been reached in the conclusions.

Good luck for your research! 

Author Response

Dear reviewer.

We would like to thank again the work performed of the reviewers and your suggestions. We have worked hard in order to improve the paper based on the comments received.

Below we explain such last corrections, point to point. First, we show the text of the reviews, while our answers appear next in red colour, in order to facilitate the checking of the corrections and the improvements in accordance to every suggestion.

  1. You should better explain the method as for the theory to be adopted. Then, in the results' section, all the numbers and data provided in  the tables have to be explained, at the least the first time a data appears. 

Thank you very much for this comment. As you request, we have clearly and explicitly included the objectives of the study and have expanded the explanation of the chosen methodology and its suitability for this research into section 2.1.

  1. Similarly, all the assumptions, limits, variables should be justified (previous research?), as well as the flowchart.

Thank you very much for this comment. Assumptions, limits and variables have been selected on the basis of the characteristics of the production process itself, since they must be adapted to the peculiarities and specificities of this process.

As you request, we have justified the selection of assumptions, limits and variables on the basis of previous research carried out by the researchers who have prepared this article.

The flowchart must be drawn up according to the reality of material, energy, water and other consumption flows, and these depend on the existing production process.

In this research, we have based ourselves on the production process studied, faithfully reflecting the reality of this process.

 We proceed to explain the origin of the flowchart in section 2.3.

  1. Finally, I'd suggest to clearly state the objective of the study at the beginning, and then summarize if and how the objective has been reached in the conclusions.

Thank you very much for this comment.

Due to the layout limitations of the document, we have been forced to divide each of the phases into 4 tables, one for each of the production processes.

However, due to the nature of the research, it is reasonable to study the data for each of the stages and processes.

As you request,, we have performed an analysis of the data shown for each of the stages as well as for the overall data.

Conclusions have also been modified again to include an explicit citation to the objectives and their achievement.

Special thanks again for your good comments, which help us to improve our article and make it a document of great scientific interest.

Round 2

Reviewer 1 Report

Dear Authors,

thanks for considering the comments from the previous review and for significantly improving the paper.

For me there are still some unanswered questions, albeit minor ones.

- Table 1, 2, 3, 4: now the data in Tables 1, 2, 3 and 4 are clearer; the only thing I don't understand is why the quantity of cork required does not appear in Phase 1 (it seems to me that only part of it is considered, e.g. 99.6 kg to produce 1000 kg of caps).

- Table 1, 2, 3, 4: due to the fact that there is a huge amount of cork waste (“almost 80% of the total mass of cork from the selected planks is not usable for the production of stoppers”), I would have expected to see this amount in Tables 1, 2, 3 and 4 in order to better define a mass balance.

- Line 237: thank you for the explanation regarding the handling of cork waste. However, in the previous review I was referring to any waste of ink, glue, washing water, etc. and wondered if and why they were excluded from the analysis.

GENERAL COMMENTS:

- I still believe that the F.U. should represent the function, so I think it is more correct to consider 1 cap, obviously accompanied by its relative geometric characteristics. In any case, the addition of Table 19 makes it possible to overcome my comment.

- With regard to cork waste, I understand that this is handled differently by each producer; I just wanted to confirm that this has been considered in the study as a by-product and what allocation has been assumed (and why, if possible).

Thank you again.

Author Response

Dear reviewer.

We would like to thank again the work performed of the reviewers and your suggestions. We have worked hard in order to improve the paper based on the comments received.

Below we explain such last corrections, point to point. First, we show the text of the reviews, while our answers appear next in red colour, in order to facilitate the checking of the corrections and the improvements in accordance to every suggestion.

Table 1, 2, 3, 4: now the data in Tables 1, 2, 3 and 4 are clearer; the only thing I don't understand is why the quantity of cork required does not appear in Phase 1 (it seems to me that only part of it is considered, e.g. 99.6 kg to produce 1000 kg of caps). - Table 1, 2, 3, 4: due to the fact that there is a huge amount of cork waste (“almost 80% of the total mass of cork from the selected planks is not usable for the production of stoppers”), I would have expected to see this amount in Tables 1, 2, 3 and 4 in order to better define a mass balance.

Thank you very much for this comment.

A criterion of material consumed in each phase has been used. In this way and in order to make it more understandable for the reader, we have considered the material consumption in each phase from an input of 1000 kg of raw cork.

That is why the net quantity obtained corresponds to the 1000 kg of raw cork minus the consumptions of each of the phases.

We proceed to include a paragraph to clarify this information.

We wish to specify that, due to the peculiarities of the manufacturing process, only a small amount of the plates that meet the geometric and organoleptic characteristics required to obtain stoppers finally become stoppers, since the rest corresponds to bark, trimmings, or remains from the polishing, cutting, etc. processes. This is why phase 2 is the phase in which the greatest consumption of raw cork is produced.

Line 237: thank you for the explanation regarding the handling of cork waste. However, in the previous review I was referring to any waste of ink, glue, washing water, etc. and wondered if and why they were excluded from the analysis.

 Thank you very much for this comment.

All consumption of ink, glue, washing water, etc. has been explicitly considered in the consumption associated with the manufacture of the stoppers.

Regarding waste treatment, wastewater is treated at the industrial park's wastewater treatment plant, while packaging and other wastes have been processed by a waste manager.

All waste has remained outside the limits of the system as indicated in the system limits section.

To clarify this point we proceed to modify Figure 2 to highlight the exclusion of waste processing

GENERAL COMMENTS:

- I still believe that the F.U. should represent the function, so I think it is more correct to consider 1 cap, obviously accompanied by its relative geometric characteristics. In any case, the addition of Table 19 makes it possible to overcome my comment.

Thank you very much for this comment.

- With regard to cork waste, I understand that this is handled differently by each producer; I just wanted to confirm that this has been considered in the study as a by-product and what allocation has been assumed (and why, if possible).

Thank you very much for this comment.

Manufacturers normally use an economic criterion for the recovery of these wastes.

Although transportation costs have a significant influence on the order of recovery of this waste, the following order of prioritization is generally followed:

  1. reprocessing of defective caps to obtain caps of other dimensions (smaller diameters, shorter lengths or caps with 1+1 or 0+1 or 0+2 washers).
  2. Colmatado of stoppers (filling of holes of the stoppers with cork powder and glue to obtain a low cost and added value stopper).
  3. Manufacture of washers from clean cork sheets obtained from the reprocessing of offcuts.
  4. Crushing of clean cork (without bark) for the manufacture of microgranulated stoppers.
  5. Crushing of dirty cork (with remains of bark) for other non-food applications.
  6. Use of shredded bark as a substrate in the agri-food sector.
  7. Use of dirty fragments and dust for the wood sector (manufacture of panels or use as biomass).

We would like to thank you for your comments and suggestions that have helped to improve the document and its scientific interest.
